# Cervical HPV Infections, Sexually Transmitted Bacterial Pathogens and Cytology Findings—A Molecular Epidemiology Study

**DOI:** 10.3390/pathogens12111347

**Published:** 2023-11-14

**Authors:** George Valasoulis, Abraham Pouliakis, Georgios Michail, Ioulia Magaliou, Christos Parthenis, Niki Margari, Christine Kottaridi, Aris Spathis, Danai Leventakou, Argyro-Ioanna Ieronimaki, Georgios Androutsopoulos, Periklis Panagopoulos, Alexandros Daponte, Sotirios Tsiodras, Ioannis G. Panayiotides

**Affiliations:** 1Department of Gynecology and Obstetrics, Medical School, University of Thessaly, 41500 Larisa, Greece; 2Department of Midwifery, School of Health Sciences, University of Western Macedonia, 50100 Kozani, Greece; 3Hellenic National Public Health Organization-ECDC, Marousi, 15123 Athens, Greece; 42nd Department of Pathology, National and Kapodistrian University of Athens Medical School, “Attikon” University Hospital, 12462 Athens, Greece; apouliak@med.uoa.gr (A.P.);; 5Department of Gynecology and Obstetrics, Medical School, University of Patras, 26504 Patras, Greece; gmichail@upatras.gr (G.M.); androutsopoulos@upatras.gr (G.A.); 63rd Department of Gynecology and Obstetrics, National and Kapodistrian University of Athens Medical School, “Attikon” University Hospital, 12462 Athens, Greece; 7Independed Researcher—Cytopathologist, Kifissias Avenue 27A’, 11523 Athens, Greece; 8Department of Genetics, Development & Molecular Biology, School of Biology, Aristotle University of Thessaloniki, 54124 Thessaloniki, Greece; 94th Department of Internal Medicine, National and Kapodistrian University of Athens Medical School, “Attikon” University Hospital, 12462 Athens, Greece

**Keywords:** epidemiology, public health, cervical screening, HPV, human papillomavirus, sexually transmitted infections, cervical pathogens, bacteria, *Chlamydia trachomatis*, *Mycoplasma genitalium*, *Mycoplasma hominis*, *Ureaplasma urealyticum*, Papanicolaou smear

## Abstract

Prevalent cervical HPV infection and high-risk HPV persistence consequences have been extensively investigated in the literature; nevertheless, any causative interrelations of other sexually transmitted bacterial infections (STIs) with cervical HPV infection have not yet been fully elucidated. This study aimed to investigate the possible association of STIs with cervical cytology aberrations and HPV genotyping results in a representative sample of predominantly young Greek women. Liquid-based cytology and molecular detection for bacterial STIs and HPV as well as extended HPV genotyping were simultaneously assessed in cervical samples from 2256 individuals visiting several urban outpatient Gynecology Departments for well-woman visits or cervical screening throughout a 20-month period. All specimens were centrally processed with validated molecular assays. The mean age of the studied women was 37.0 ± 11.7 years; 722 women (33.30%) tested positive for STI (mean age 34.23 ± 10.87 years). A higher mean age (38.34 ± 11.83 years (*p* < 0.05)) was associated with negative STI testing. *Chlamydia trachomatis* was detected in 59 individuals (8.2%), *Mycoplasma hominis* in 156 (21.6%), *Mycoplasma genitalium* in 14 (1.9%), and *Ureaplasma* spp. in 555 (76.9%); infections with two bacterial pathogens were identified in 73 samples (10.1%). Cervical HPV was detected in 357 out of 1385 samples with a valid HPV typing result (25.8%). The mean age of HPV-positive women was 32.0 ± 8.4 years; individuals testing HPV-negative were slightly older (N = 1028): 34.4 ± 9.2 (*p* < 0.05). Among the 1371 individuals with valid results both for bacterial STIs and cervical HPV detection, women with an HPV-positive sample were more likely to harbor an STI (OR: 2.69, 95% CI 2.10–3.46, *p* < 0.05). Interestingly, bacterial STI positivity illustrated significant heterogeneity between NILM and LSIL cases, with 28.88% of NILM and 46.33% of LSIL cases harboring an STI, respectively (*p* < 0.05). In brief, in a population with a high prevalence for STIs, especially *Ureaplasma* spp., an association was documented between bacterial pathogen detection and cervical HPV infection, as well as abnormal cytology; these findings merit further investigation.

## 1. Introduction

With their long-term reproductive system complications (cervicitis, endometritis, tubal-factor infertility, pelvic inflammatory disease, susceptibility to ectopic pregnancy, and HIV acquisition), sexually transmitted infections (STIs) cause significant morbidity in sexually active women, especially adolescents and young nulliparous women, therefore representing a major global health priority [1,2]. These infections are often diagnosed simultaneously with human papillomavirus (HPV) infections, the latter being the most prevalent sexually transmitted disease (STD) among women aged <35 years worldwide [3,4]. Lately, the identification of HPV as the etiologic factor of anogenital tract precancer and neoplasias has dramatically changed current clinical practice [5,6]. Among the more than 180 HPV genotypes identified so far, predominantly 14 cause persistent infections and, thus, are classified as high risk (HR-HPVs) based on their oncogenic potential, while the remainder are considered as intermediate, low, or of uncertain risk, reflecting their association with cervical cancer (CC) development [7,8,9]. While the majority of HPV infections based on their natural history are mostly transient and spontaneously regress within a short period of time (12 to 24 months), HR-HPV persistence and latency are the main causative factors associated with high-grade precancerous cervical lesions (cervical intraepithelial neoplasia—CIN2-3/high-grade squamous intraepithelial lesions—HSIL), as well as invasive CC development [10,11,12].

Despite the extensive literature focusing on the consequences of cervical HR-HPV infection, as well as guidelines on STI management that are updated periodically by global stakeholders (e.g., WHO, CDC), the possible interrelationship of other sexually transmitted infections detected concurrently with cervical HPV infection have not been fully elucidated yet [13,14,15,16]. Novel, widely available molecular assays offer the opportunities for accurate simultaneous detection of multiple pathogens in cervical secretions. Especially for young cohorts with suboptimal HPV vaccination rates, defining the epidemiology of cervical co-infections (concurrent isolation of HPV and bacterial STIs) is crucial both for avoiding cervical precancer overtreatment as well as preventing antibiotic resistance.

This study aimed to investigate the possible association of STIs with cervical cytology aberrations and HPV genotyping results in a large representative sample of predominantly young Greek women.

## 2. Materials and Methods

### 2.1. Study Population—Inclusion and Exclusion Criteria

We conducted a prospective pragmatic (real world) observational study enrolling eligible women who attended outpatient gynecology departments for general gynecological examination or routine cervical screening or colposcopy clinics of urban university hospitals located in 3 different Greek cities (Athens/Larisa/Patras). The study was run throughout a 20-month period (between October 2015 and June 2017). Following detailed briefing regarding the scopes of the study, a Thin Prep cervical sample was obtained from individuals who agreed and signed the informed consent form. Women who were pregnant at the time of enrolment, had any immunosuppressive condition, those who had been previously reviewed in colposcopy for abnormal cytology, or had prior ablative or destructive treatment of cervical precancerous lesions were excluded. We also excluded women who reported ever receiving treatment for any bacterial STI in the past.

### 2.2. Study Protocol

According to the study protocol, detailed history (medical and gynecological) was obtained from all eligible individuals at the first visit, covering aspects such as age at first pregnancy, parity, number of lifetime sexual partners, recent changes in sexual partners, condom use, and HPV immunization. All women were informed about the scope of the study and were asked to sign a consent form at enrollment. In addition to epidemiological data, other confounding factors affecting HPV status and concurrent CIN (such as smoking) were documented. Consequently, liquid-based cytology (LBC) samples were obtained from all participants using Rovers™ Cervex brushes and transferred into PreservCyt solution. This was followed by centrally performed (Attikon University Hospital) cytological and biomolecular analysis of HPV DNA and other bacterial STIs (*Chlamydia trachomatis* spp. (CT), *Mycoplasma hominis* spp. (MH), *Mycoplasma genitalium* spp. (MG), and *Ureaplasma urealyticum/parvum* spp. (UU/UP)). For these scopes, two validated assays have been utilized:HPV DNA Genotyping (CLART-2 HPV Test^®^ (Genomica, Madrid, Spain)).CLART^®^ STIs kits (GENOMICA, Madrid, Spain).

When an STI or cervical pathology was identified, individuals were referred for appropriate management, outside the context of this study.

The study was conducted within a national multidisciplinary research protocol in cervical pathology and was approved by the Greek Central Government (Ministry of Education and Religious Affairs) under the framework and funding of the HPVGuard research project (http://HPVGuard.org, Project Number: 11ΣΥΝ_10_250, Cooperation framework, Protocol Number: ΕΥΔΕ—ΕΤAΚ 1788/1-10-2012) and subsequently received additional approval from the co-ordinating authority—the Attikon University Hospital Ethics Committee (code: ΕΒΔ 623/14-5-13).

In this manuscript, we are presenting datasets from eligible individuals for whom STI assay results as well as LBC and/or HPV genotyping results were available. In order to obtain a more robust dataset, we did not exclude cases for which some individual exam results were missing (for example, due to insufficient biological material due to prior consumption for other examinations or failure of the molecular tests); these cases were used in pairs for those parts of the analysis where results were available and valid statistical tests could be obtained (refer to Figure 1 for a Venn diagram indicating the examinations presenting simultaneously valid results).

Ectocervical and endocervical samples, using the ThinPrep^®^ Pap test, were collected by Rovers™ Cervex brushes and transferred into PreservCyt solution. PreservCyt^®^ vials (Cytyc Inc., Boxborough, MA, USA) containing the cellular material were used to prepare mono-layer slides using the ThinPrep^®^ 2000 Automated Slide Processor^®^ (Cytyc, Boxborough, MA, USA) according to the manufacturer’s instructions. Cytological results were expressed according to the Bethesda classification system (8): (i) negative for intraepithelial lesion or malignancy (NILM); (ii) atypical squamous cells of undetermined significance (ASC-US); (iii) low-grade squamous intraepithelial lesion (LSIL); (iv) high-grade squamous intraepithelial lesion (HSIL); (v) squamous cell carcinoma (SCC); and (vi) adeno-carcinoma (AdenoCa).

HPV DNA detection was performed on the same biological material implementing a validated molecular assay (CLART2 HPV, Genomica, Coslada, Madrid, Spain) capable of identifying individually the 35 most common HPV genotypes, both HR-HPVs as well as several low-risk HPVs (LR-HPVs) (6, 11, 16, 18, 26, 31, 33, 35, 39, 40, 42, 43, 44, 45, 51, 52, 53, 54, 56, 58, 59, 61, 62, 66, 68, 70, 71, 72, 73, 81, 82, 83, 84, 85, and 89).

The samples were tested for bacterial STIs using a DNA microarray system for the detection and molecular identification of pathogens causing sexually transmitted infections. The microorganisms detected were the following: *Chlamydia trachomatis* (CT), *Mycoplasma hominis* (MH), *Mycoplasma genitalium* (MG), and *Ureaplasma urealyticum/parvum* (UU/UP). DNA was extracted, amplified, and analyzed for the detection and genetic identification of pathogens causing sexually transmitted infections using microarray methods and CLART^®^ STIs kits according to the manufacturer’s instructions (GENOMICA, Madrid, Spain). Briefly, 1 mL of the homogenized sample was placed in a sterile 1.5 mL microcentrifuge tube and centrifuged at 12,000 rpm for 10 min to obtain the pelleted form. Subsequently, the supernatant was discarded; the pellet was resuspended in 1 mL sterile water and centrifuged at 12,000 rpm for 10 min. The DNA extraction procedure was followed by the addition of 180 μL lysis buffer T1 and 25 μL proteinase K to the pellet and incubation of the samples in a thermostatic mixer at 56 °C and 550 rpm for 1 h. At the end of the DNA extraction method, 100 µL of eluted DNA was recovered and stored at −20 °C.

We analyzed cytology results in the aforementioned separate Bethesda-based categories (NILM, ASC-US, LSIL, and HSIL) [17] and correlated them with HPV status and bacterial STI detection results.

The statistical analysis platform used for all tests was based on SAS version 9.4 for Windows software (SAS Institute Inc., Cary, NC, USA) and a *p* value of <0.05 was considered statistically significant. The t-test was used to evaluate the statistical significance for continuous variables (such as patient age) in cases where two groups were considered and data normality was assured. When more than two groups were compared and data normality was also assured, we implemented the ANOVA test and the t-test was used for post hoc analysis. Data were tested for normality by the Kolmogorov–Smirnov test and, when normality was not ensured, the Mann–Whitney U test or the Kruskal–Wallis test was applied (for two or more groups, respectively). For comparison of categorical variables, the χ^2^ test was performed and odds ratios were calculated whenever appropriate, i.e., for 2 × 2 contingency tables. All tests were two-sided.

## 3. Results

In total, 2256 cases were found eligible for further analysis (mean age: 36.98 ± 11.65 years, min = 18, max = 75), since they possessed a valid result in at least one of the three studied tests.

### 3.1. Analysis of STI Detection Results

One-hundred cases either had invalid results or the biological material was insufficient for adequate testing. Out of all cases with a valid STI detection result, 722 (33.30%) tested bacterial-STI-positive. The mean age of positive cases was 34.23 ± 10.87 yrs and women negative for STIs were about 4 years older, with a mean age of 38.34 ± 11.83 yrs (*p* < 0.05). Out of the 722 positive cases, 59 (8.17%) had CT, 156 (21.61%) MH, 14 (1.94%) MG, and 555 (76.87%) UU/UP. Moreover, 73 (10.11%) women suffered from dual bacterial infections; triple or higher order infections were not detected in the study population.

### 3.2. Analysis of HPV Typing Results

Out of 1385 samples with valid HPV typing results (see Figure 1), 357 (25.78%) were HPV-positive. The mean age of HPV-positive women was 32.03 ± 8.41 yrs; HPV-negative women (N = 1028) were approximately two years older: 34.46 ± 9.20 (*t*-test: *p* < 0.05). Out of the 35 detectable subtypes, 2 LR-HPV genotypes (26 and 71) were not detected in this cohort; from the remaining, the most frequent were 16 (16.53% of the positive women), 31 (15.97%), 51 (13.45%), 66 (13.17%), 53 (11.00%), and 6 (10.67%) (see Table 1 for a complete list of HPV genotype frequencies). Among the 357 patients with a documented HPV infection, 223 (62.46%) harbored a single genotype, while, in 90 (25.21%), two HPV subtypes were isolated; in all other cases, multiple genotypes were documented. HR-HPV types were confirmed in 298 out of 357 (83.47%) HPV-positive women and LR-HPV types in 101, respectively (28.29%) (see Table 2 for details related to the number of genotypes found in the studied population).

### 3.3. Liquid-Based Cytology (LBC)/Test Papanicolaou Outcomes

Out of the 1806 cases with available and valid cytology results, 1373 (76.02%) corresponded to NILM with a mean age of 38.83 ± 12.52 yrs, 86 (4.76%) were diagnosed as ASC-US with a mean age of 32.52 ± 8.52 yrs, 339 (18.77%) were categorized as LSIL with a mean age of 32.07 ± 8.10 yrs, and 7 (0.39%) corresponded to HSIL with a mean age of 40.00 ± 12.88 yrs. Finally, one single SCC case, aged 59, was documented (0.06%) (See Figure 2). Differences in the ages among the various cervical cytologicaldiagnostic categories were statistically significant (*p* < 0.001).

### 3.4. Analysis of STIs and HPV for the Age Groups

Focusing on the subgroup of the 58 women of the youngest age (≤20 yrs) for whom a valid STI and an HPV test were both available, 23 (39.66%) had a bacterial STI detected. There were no significant differences in the proportions of infected individuals between the ≤20 yrs group and the 21–30 yrs group (*p* = 0.9356). As for STI positivity, this was lower in the older age groups; specifically, 33.33% in the 31–40 group, 21.8% in 41–50 group, 26.19% in the 51–60 age group, and, finally, 12.5% in the senior, 61–70 age group (see Table 3). Differences in STI positivity rates were statistically significant among the various age groups (chi-square *p* < 0.001). Similarly, HPV positivity percentages were waning in the older age groups (Table 3).

Coinfections (HPV and bacterial STI) were mostly prevalent (22.41%) in the youngest age group (≤20 yrs) and less common in older age groups, specifically 15.9% (21–30 yrs), 12.07% (31–40 yrs), 8.65% (41–50 yrs), and 7.14% (in the 51–60 yrs); indeed, coinfections were not documented in ages 60+ (Table 3). HPV-positive/STI-negative women were mostly seen in the young age groups 21–30 yrs (14.70%) and 31–40 yrs (12.97%). HR-HPVs were more common in the younger age groups of ≤20 and 21–30 yrs (Table 3). The prevalence of *Ureaplasma* spp. (UU/UP) gradually decreased with advancing age (Table 3). However, considerable MH prevalence (11.90%) was observed in the 51–60 age group, comparable to that of younger age groups ≤ 20 (13.79) and 21–30 (8.43%).

### 3.5. Analysis of STIs and HPV for the Cervical Cytology According to the Bethesda Classification Groups

Bacterial STI positivity differed significantly between NILM and LSIL cases, with 28.88% of the NILM cases harboring an STI, compared to 46.33% of the LSIL cases (*p* < 0.05) (Table 4). Comparable infection rates were observed among ASC-US, LSIL, and HSIL cases (Table 4, 33.33%, 46.33%, and 33.33%, respectively, without statistically significant difference in pairs). As anticipated, significant differences were documented in the HPV infection status between NILM and LSIL, NILM and HSIL, ASC-US and LSIL, as well as ASC-US and HSIL (in all cases corresponding to *p* < 0.05). For the sub-population of HPV-positive cases, no significant differences in the STI status were observed among the cytological subcategories; however, for the HPV-negative sub-group, bacterial STI positivity rate was higher in the LSIL group than in NILM cases (OR: 0.57, 95% CI: 0.39–0.82, *p* = 0.0026, see Table 4).

In relation to the risk for an abnormal cytological outcome, predictably, irrelevant of the STI status, HPV-positive cases presented higher chances for an abnormal cytology than the corresponding HPV-negative (OR: 4.99, 95% CI 3.78–6.59, *p* < 0.05), while this risk decreased in STI-negative women (OR: 3.37, 95% CI 2.62–4.35, *p* < 0.05). In cases with unknown/unavailable HPV status, cases testing positive for bacterial STI were linked with abnormal cytology (OR: 1.89, 95% CI 1.47–2.43, *p* < 0.05). Women testing positive for both an STI and HPV had a higher chance of concurrent cytological abnormalities (n = 57/800 = 7.13% in the NILM group and n = 96/397 = 24.18% in the cytological abnormal group, Table 4), OR: 4.16, 95% CI 2.92–5.90, *p* < 0.05), while patients negative for both HPV and a bacterial STI illustrated higher chances of normal cytology than cases harboring either or both (bacterial and HPV) infection types (OR: 3.44, 95% CI 2.67–4.43, *p* <0.05, Table 4).

Among HPV-positive individuals with valid cytology and STI outcomes, 121 out of 309 (39.16%) had NILM cytology. ASC-US was identified in 21 (6.78%), LSIL was detected in 162 out of 309 (52.43%), and HSIL in 5 (1.62%) (Table 4). Out of the 888 HPV-negative patients, 679 (76.46%) had normal cytology, 57 (6.42%) were categorized as ASC-US, 151 (17%) LSIL, and 1 (1.13%) corresponded to HSIL; the sole SCC case was HPV-positive.

Out of the cases with valid STI results as well as HPV detection outcomes (n = 1371), CT was detected in 35 (7.61%) patients, MH in 99 (21.52%), MG in 11 patients (2.39%), while UU/UP was detected in 350 (76.09%) patients (Appendix A). HPV-positive women were more likely to harbor a bacterial STI (OR: 2.69, 95% CI 2.10–3.46, *p* < 0.05); a similar situation was observed when we studied women with HR-HPV infection (irrelevant of the LR-HPV infections) or LR infections (irrelevant of HR infections) (see Appendix A). Interestingly, women that harboured both HR- as well as LR-HPV genotypes had four-times higher chances of testing positive for a bacterial STI than women with either an HR- or an LR-HPV genotype (OR: 4.15, 95% CI 2.16–7.97, *p* < 0.05).

Out of the 1371 cases with valid results for HPV and STIs, in 35 individuals, CT was detected (2.55%), which illustrated the strongest correlation with HPV positivity (OR: 3.19, 95% CI 1.62–6.26, *p* < 0.05) and a stronger correlation with LR- than HR-HPVs (OR: 4.01, 95% CI 1.77–9.07, *p* < 0.05 and OR: 2.53, 95% CI 1.27–5.03, *p* < 0.05, respectively, see Appendix A) and, additionally, higher risk if both LR- and HR-HPVs were present (OR: 4.41, 95% CI 1.48–13.11, *p* < 0.05). Ninety-nine women (7.22%) were positive for MH; the odds ratio of its association with HPV ranged between 1.95 and 2.45 (see Appendix A). MG was detected in a very small percentage of women (0.8%) precluding safe conclusions; however, it was detected in comperable proportions in HPV-negative cases as well as HR-, LR-, and both HR- and LR- (see Appendix A) and seemed mostly associated with HPV-negative cases.

Finally, 350 (25.23%) women tested positive for UU/UP; for these women, the risk of UU/UP positivity increased as the infection switched from LR-HPV-related (OR: 1.97, 95% CI 1.29–3.01, *p* < 0.05) to HR-HPV-related (OR 2.41. 95% CI 1.83–3.17, *p* < 0.05) or simultaneously LR- and HR-HPV-related (OR: 3.04, 95% CI 1.64–5.64, *p* < 0.05) (see Appendix A).

## 4. Discussion

In this study, in a European population with relatively high bacterial STI prevalence (*Ureaplasma* Spp. in particular), illustrating cervical HPV prevalence and genotype distribution consistent with previously reported epidemiology in a national setting, we have documented an association between bacterial pathogen detection and HPV infection as well as abnormal cervical cytology [18,19]. Of particular interest is this study’s finding that women testing positive both for HR- as well as LR-HPV genotypes illustrated a relative risk (RR) of 4 to also test positive for a bacterial STI when compared to individuals testing positive individually either for an HR- or an LR-HPV genotype in isolation.

### 4.1. Study’s Findings in the Regional Context

In this multicenter molecular epidemiology study, which recruited representative cohorts of reproductive-age women, the largest materialized in Greece so far, simultaneous infections by bacterial STIs together with cervical HPV have been frequently detected. The distribution of HPV genotypes in this large multicenter cohort is consistent with the findings of previously reported studies conducted by our group [18]. *Ureaplasma* spp. was the most prevalent *Mycoplasmataceae* detected, followed by MH and CT. This is in line with findings from previous smaller scale, similarly designed single-center studies which have been conducted regionally during the past decades [20,21,22,23,24]. *Ureaplasma urealyticum* is a bacterium belonging to the genus *Ureaplasma* and the family *Mycoplasmataceae* in the order *Mycoplasmatales*, representing one of the smallest cellular microorganisms found in nature [25]. It can be isolated in the urogenital system of many healthy individuals as a commensal. In 2002, *Ureaplasma urealyticum* was divided into two species, *U. parvum* (UP) (biotype 1) und *U. urealyticum* (UU) (biotype 2) [26].

Despite the absence of formal nationwide data, the literature clearly suggests that *Ureaplasma* spp. is endemic in Greece, as is the case with other geographical areas. In this perspective, our study’s findings might reflect regional bacterial STI variability. In a previous study also focusing on the possible association of HPV and STI codetection with cytological findings, Parthenis et al. recruited prospectively 345 asymptomatic patients attending a Greek urban gynecology clinic for routine cervical screening [27]. In this cohort, *Ureaplasma* spp., detected in 18.2% of participants, was the most frequently isolated pathogen; one in every four women in this study testing positive for this bacterium additionally harbored an HR-HPV genotype. In another contemporary Greek study, comprised of 347 asymptomatic women undergoing routine cervical screening in an urban setting, 16.13% of the studied individuals carried *Ureaplasma* spp. (predominantly UP) in high concentrations [28].

In a different setting, that of an urban Greek STD outpatient clinic, Mortaki et al. conducted a cross-sectional study in which cervicovaginal smears of women with anogenital warts were examined for the presence of HR-HPV types and common STIs. In contrast with CT coinfection rates, which were similar across the study groups, the authors report that coinfections with *Ureaplasma* spp., MH, and MG were more common in patients with warts, with 45.9% of individuals among this cohort being diagnosed with *Ureaplasma* spp. [29]. Finally, in the context of unexplained chronic voiding symptoms, a high prevalence of UU was detected among 153 Greek women [21].

Other studies from neighboring countries have also investigated the concurrent detection of cervical HPV together with bacterial STIs. In a recent Italian study, Martinelli et al. documented a relatively high percentage of women with CT infection, alone or in combination with seven HR-HPV types, in individuals attending gynecology outpatient clinic following an abnormal Pap smear [30]. Another recent Italian study also underscored the need for surveillance to implement tailored vaccination programs and cervical cancer preventive strategies [31].

### 4.2. Bacterial STIs and Ureaplasma spp. in Particular as a Potential Co-Factor in Cervical Carcinogenesis

There is ample literature evidence that, in addition to the key role of HR-HPVs, cervical carcinogenesis is also associated with inflammation [32,33]. Interestingly, by obscuring the cytologic identification of atypical cells, persistent cervicitis might also enhance the progress of undetected precancerous cervical lesions [32]. Cervicitis has been associated with a loss of cervical columnar cells, a typical feature of the maturational process; thus, STIs might represent squamous metaplasia promoters [34]. Perhaps the initial steps of HPV-mediated carcinogenesis are helped by a state of cervical inflammation, driven predominantly by the hormonal milieu, regulatory cytokines and chemokines, as well as multiple cervicovaginal microorganisms [32,35,36].

Among bacterial STIs, the detrimental effects of *Chlamydia trachomatis* (CT) cervical infection have been suspected for decades; currently, the association between CT and CC has been well established [23,37,38,39,40,41]. Hypothetically, CT might increase susceptibility to HPV causing microabrasions or cervical epithelial cells and molecular alterations, thus facilitating the entry of virions [23,37,42]. In a recent systematic review and meta-analysis of 48 studies assessing the possible association between HPV and CT infection, Naldini et al. documented that, among women harboring CT, the odds ratio (OR) of HR-HPV infection was 2.32 (95% CI 2.02, 2.65), while the OR for CT among HPV-positive women was 2.23 (95% CI 1.70, 2.92). The authors consider HPV and CT behaving as reciprocal risk factors, concluding that, in women diagnosed with either cervical HPV or CT, screening for the mutual infections represents a justified preventive intervention both for CC as well as infertility [39]. In the interaction between CT and HPV, several host modulating factors (genetic background, endogenous hormones, and immune response variations) might also be shared with other bacterial STIs [36].

*Ureaplasma* spp. are common STI pathogens frequently found in the healthy female genitourinary tract; therefore, their pathogenic role in individuals is difficult to substantiate [24,43]. In an early study, Lukic et al. postulated that UU is related to the persistence of HPV infection and early cervical cytological changes [35]. Drago et al. suggested that UP may be involved in the carcinogenic process of HPV, directly influencing the expression of HPV proteins or indirectly by stimulating a persistent inflammatory process [44]. The chronic inflammation caused by *Ureaplasma* spp. infections might favor the entry of other microorganisms, act as cofactor in the pathogenesis of cervical disease, or induce chromosomal alterations that might lead to carcinogenesis of epithelial cells [45]. The possible mechanism of the association between UU infection and abnormal cervical cytopathology might be related to the combination of several complex infection-associated inflammatory responses, involving production of reactive oxidative metabolites, increased expression of cytokines, chemokines and growth and angiogenic factors, decreased cell-mediated immunity, and the generation of free radicals [40,46]. The large meta-analysis of Liang et al. concluded that, together with bacterial vaginosis (BV), CT, and reduction in Lactobacilli, UU are also associated with increased risk of HPV infection and CIN development [40,47,48].

From a microbiome perspective, *Ureaplasma* spp. vaginal colonization at low levels is seemingly harmless [49]. In the study of Veteramo et al., a significant association between HPV and UU was documented, however, only at high-density colonization rates (HDC-UU), leading the authors to consider that UU pathogenic potential only emerged in high densities [23]. Similarly, in the study of Kim et al., only HDC-UU was significantly associated with HPV infection. Interpreting their results, the authors suggest that, at HDC-UU rates, even asymptomatic UU infection should be eradicated, regardless of age, for the prevention of HPV infection and subsequent CIN [50]. This is consistent with the opinion that using *Ureaplasma* bacterial load as a diagnostic criterion might be required to decide on appropriate drug intervention [51,52]. In our study, the specifications of the preselected assay as well as the high inflicted costs precluded the measurement of density colonization rates in UU-positive cases. Further studies will be required to confirm whether there are “safe levels” for *Ureaplasma*, as is also the case for MH [53].

The study of Veteramo et al. also evidenced the lack of protective precautions against STIs, both in HPV-positive and HPV-negative women. Despite the debate on the effectiveness of condoms in reducing HPV transmission rates, counseling on STI prevention use should be continued to prevent bacterial pathogen transmission [23,54].

### 4.3. Cost Effectiveness Considerations

With their range of severe negative adverse reproductive consequences, early detection and treatment of bacterial STIs in the susceptible population seems all-important. However, several factors contribute in suboptimal STI control globally (their relapsing nature, unreliable assays, limited access to health facilities, inadequate infrastructures, possible effects of sexual networks, immunosuppression, etc.) [55,56,57].

Interestingly, there is currently no unanimity in viewpoints and guidelines for STI detection screening policies [58]. A position statement recently developed by the European STI Guidelines Editorial Board advises against routine MH, UU, and UP testing and treatment, not only for asymptomatic but for symptomatic women as well. The authors argue that asymptomatic bacterial carriage is common and the majority of individuals will not develop related disease. Based on the position statement, extensive testing, detection, and antimicrobial treatment of these bacteria might ultimately result in the selection of antimicrobial resistance towards more “aggressive” STIs as well as in the general microbiota and substantial economic cost for societies and individuals. The authors consider that the recent “commercialization” of several multiplex PCR assays detecting typical nonviral STIs together with MH, UU, and UP has worsened this situation [59].

Undeniably, cost-effectiveness appraisal of bacterial STI screening policies is multi-factorial, linked with several mid- and long-term social and public health correlates, frequently disregarded by mathematic modeling. Certainly, reliable diagnosis by the novel multiplex RT PCR assays offering simultaneous detection of cervical pathogens largely facilitates their management nowadays [8,60,61]. Furthermore, there is already sufficient evidence that, by complicating the natural course of cervical HPV infections, bacterial STIs prolong HPV clearance, thus leading to complex morbidity and excess economic burden [18,62].

### 4.4. Strengths and Limitations of This Study

With 2256 individuals enrolled in the analysis, this study’s major strength is predominantly the sizable patient sample. Our study represents the largest relevant prospective work conducted in Greece so far. Furthermore, all cytological and molecular assessments (both for bacterial STIs as well as HPV) have been performed in a single university-affiliated laboratory (thus minimizing variability), implementing state-of-the-art molecular assays and undergoing regular external QA. The extended HPV genotyping assay utilized in this study allows for the detection of subtle differences in the interaction of specific bacterial STIs with either LR-HPVs or HR-HPVs in a subsequent work.

The main limitation of this pragmatic (real-world) study is the incomplete dataset of assays affecting several individuals, with missing results either for cytology, a particular bacterial STI, or for HPV assessment (Figure 1). Further, we did not examine the presence of pathogens like *Trichomonas* spp. Or *Candida* that may have an additional role either as true pathogens, facilitators, or commensals [40,63]. After performing an evaluation of the effect of imputations on the final estimations, we bypassed this issue as described in the Material and Methods section. Second, because of the study’s protocol, no data on histology, which represents the “gold standard”, were available; obtaining cervical biopsies for mostly benign conditions would cause unjustified and unnecessary iatrogenic morbidity. Third, since a national vaccination registry has only recently been introduced in Greece, after documenting inaccuracies in the self-reported HPV vaccination questionnaire data, we chose to omit any sub-analysis comparing HPV vaccinated with non-vaccinated individuals [64]. Fourth, since consistent condom use (>90% of times) was reported by very few (<5%) individuals, we decided against embarking on a sub-analysis to evaluate condom effect on the distribution of HPV or bacterial STI positivity rates [18,54]. Fifth, despite smoking’s established role in cervical carcinogenesis, with several individuals (smokers and non-smokers) reporting use of alternative simulating devices (vaping or e-cigarette), smoking exposure data were uncertain to quantify and were omitted. Finally, as stated, no sub-analysis was included comparing the effect of either LR-HPVs or HR-HPVs covariation with cytology and bacterial STI expression.

## 5. Conclusions

With a host of complex interrelated mechanisms, HPV and bacterial STIs cause detrimental effects on female fertility as well as significant psychosomatic burden imposed by the disease and the related treatments. Several physicians and authors indeed consider their codetection an anticipated finding, since both STIs and HPV represent sexual exposure correlates [65]. Most likely, only future in-depth in vitro studies will ultimately confirm whether *Ureaplasma* spp. and/or other STIs are real cofactors or are just “followers”, taking advantage of the immune tolerance and abnormal regulation of the cell cycle control generated by HPV for the high prevalence of STIs found in HR-HPV-positive women [66].

With vaccinated cohorts gradually entering cervical screening, future studies will investigate the long-term public health effects of HPV vaccination on bacterial STI prevalence at the population level [67]. Despite the elusive underlying molecular mechanisms, the variability in guidelines, and a questionable cost-effectiveness profile, we consider that screening for bacterial STIs should be encouraged, at least for reproductive-age women harboring cervical HR-HPV or CIN. In the near future, the potential coadministration of the HPV vaccine together with anti-STI vaccines currently under development might emerge as a cost-effective strategy [68]. As for now, the feasibility of HPV status, bacterial STI, and cytology coassessment in vaginal self-sampling material is particularly attractive and potentially more cost-effective [61,69].

## Figures and Tables

**Figure 1 pathogens-12-01347-f001:**
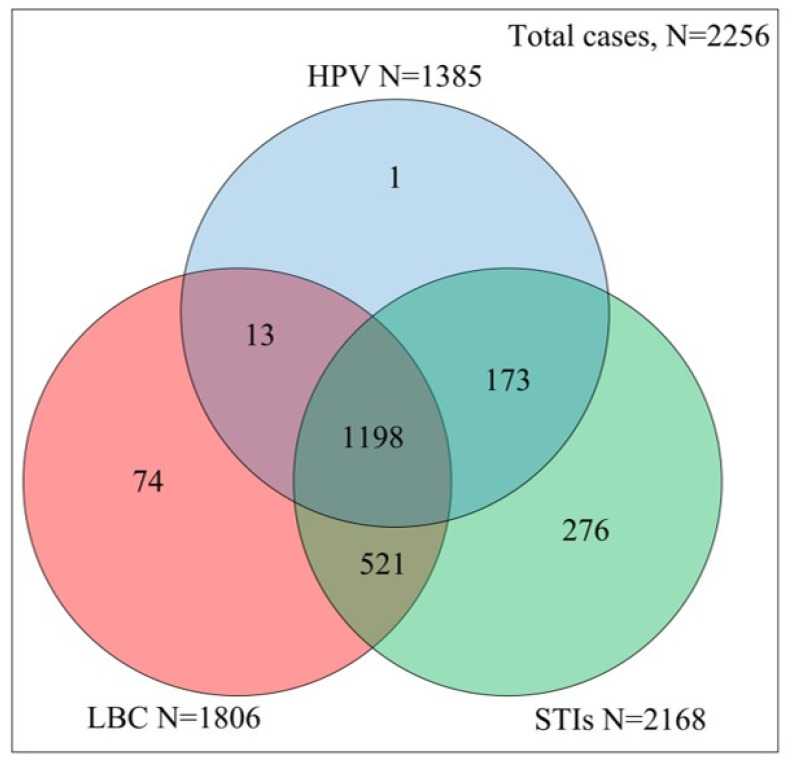
Venn diagram indicating the number of available valid data for each examination type. The total number of cases that had at least one valid outcome from HPV, STIs, and LBC were 2256. The number of women with valid HPV, LBC, and STI test were 1385, 1806, and 2168, respectively. Intersection of two circles shows the number of valid results for both sets, for instance, simultaneous HPV and STI test was available for 1371 women, while, for HPV and STIs and LBC, for 1198 women.

**Figure 2 pathogens-12-01347-f002:**
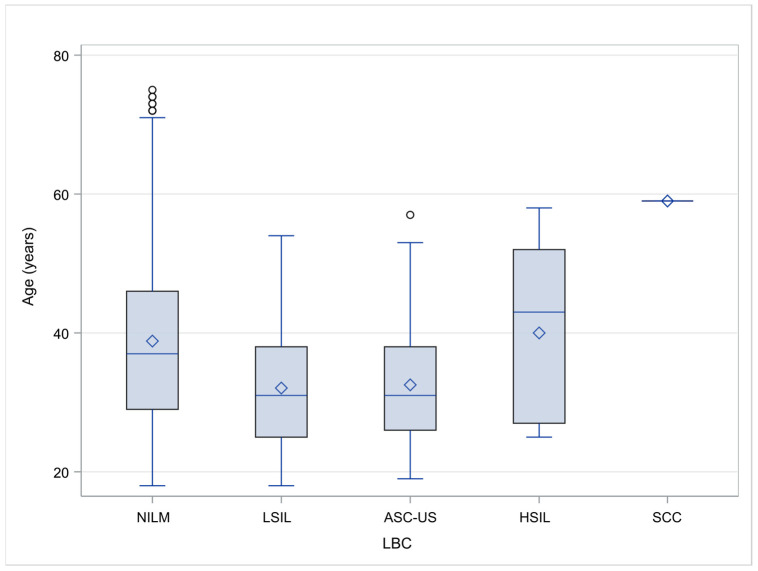
Box and whisker plot of women’s ages for each cytological diagnostic category.

**Table 1 pathogens-12-01347-t001:** Frequency of HPV subtypes in the HPV-positive population.

HPV Subtype	Subtype Prevalence in the Positive Population	HPV Subtype	Subtype Prevalence in the Positive Population
16	16.53%	84	3.00%
31	15.97%	54	2.33%
51	13.45%	83	2.33%
66	13.17%	45	1.40%
53	11.00%	62	1.33%
6	10.67%	11	1.00%
52	7.56%	82	1.00%
59	7.00%	40	0.67%
58	6.72%	44	0.67%
70	5.67%	73	0.67%
18	5.60%	85	0.67%
35	5.32%	43	0.33%
68	5.32%	72	0.33%
39	4.48%	89	0.33%
61	4.33%	26	0.00%
42	4.00%	71	0.00%
56	3.92%	HR	83.47%
81	3.67%	LR	28.29%
33	3.08%		

**Table 2 pathogens-12-01347-t002:** Number of women harboring any, HR, and LR subtypes and percentage in relation to the number of simultaneous HPV subtypes (only for the HPV-positive population).

	HPV Subtypes		Number of HR Subtypes		Number of LR Subtypes	
Number of Subtypes	N	%	N	%	N	%
0	NA	NA	59	16.53	256	71.71
1	223	62.46	203	56.86	82	22.97
2	90	25.21	66	18.49	17	4.76
3	32	8.96	22	6.16	2	0.56
4	8	2.24	5	1.40		
5	2	0.56	2	0.56		
6	2	0.56				

**Table 3 pathogens-12-01347-t003:** Distribution of HPV status and STIs for the studied population per age group.

		Age Group *					
Status **	Total n = 1344	< = 20 n = 58	21–30 n = 415	31–40 n = 555	41–50 n = 266	51–60 n = 42	61–70 n = 8
STI (+) n (%)	449 (33.41)	23 (39.66)	171 (41.2) ^a^	185 (33.33) ^b^	58 (21.8) ^a,b^	11 (26.19)	1 (12.5)
STI (−) n (%)	895 (66.59)	35 (60.34)	244 (58.8)	370 (66.67) ^c^	208 (78.2)	31 (73.81)	7 (87.5)
HPV (+) n (%)	339 (25.22)	17 (29.31)	127 (30.6) ^c^	139 (25.05) ^c^	48 (18.05)	7 (16.67)	1 (12.5)
HPV (−) n (%)	1005 (74.78)	41 (70.69)	288 (69.4)	416 (74.95)	218 (81.95)	35 (83.33)	7 (87.5)
HPV+ STI+ n (%)	172 (12.8)	13 (22.41)	66 (15.9)	67 (12.07)	23 (8.65)	3 (7.14)	0 (0)
HPV+ STI− n (%)	167 (12.43)	4 (6.9)	61 (14.7)	72 (12.97)	25 (9.4)	4 (9.52)	1 (12.5)
HPV− STI+ n (%)	277 (20.61)	10 (17.24)	105 (25.3) ^d^	118 (21.26) ^e^	35 (13.16 ) ^d,e^	8 (19.05)	1 (12.5)
HPV− STI− n (%)	728 (54.17)	31 (53.45)	183 (44.1)	298 (53.69)	183 (68.8)	27 (64.29)	6 (75)
HPV HR + n (%)	283 (21.06)	15 (25.86) ^f^	110 (26.51) ^g,h,i^	114 (20.54) ^g,j^	39 (14.66) ^f,h,j^	5 (11.9) ^i^	0 (0)
HPV HR− n (%)	1061 (78.94)	43 (74.14)	305 (73.49)	441 (79.46)	227 (85.34)	37 (88.1)	8 (100)
*Chlamydia T*. n (%)	33 (2.46)	2 (3.45)	19 (4.58)	12 (2.16)	0 (0)	0 (0)	0 (0)
*Mycoplasma H*. n (%)	96 (7.14)	8 (13.79)	35 (8.43)	35 (6.31)	13 (4.89)	5 (11.9)	0 (0)
*Mycoplasma G*. n (%)	11 (0.82)	1 (1.72)	5 (1.2)	4 (0.72)	1 (0.38)	0 (0)	0 (0)
*Ureaplasma* spp. n (%)	342 (25.45)	12 (20.69)	129 (31.08)	145 (26.13)	47 (17.67)	8 (19.05)	1 (12.5)

* Superscripts indicate pairs with *p* < 0.05; ** percentages in parentheses denote the percentage of the population within the age group that is compliant with the denoted status. Details of the statistical tests for the pairs that exhibited observed difference: ^a^: OR: 0.47, 95% CI: 0.36–0.62. *p* < 0.0001, ^b^: OR: 0.48, 95% CI: 0.28*0.83, *p* = 0.0069, ^c^: OR: 13.57, 95% CI: 1.50–123.04, *p* = 0.010 (Fisher exact), ^d^: OR: 0.17, 95% CI: 0.12–0.22, *p* < 0.0001, ^e^: OR: 2.91, 95% CI: 1.68–5.03, *p* < 0.0001, ^f^: OR: 0.57, 95% CI: 0.39–0.82, *p* = 0.0026, ^g^: OR: 0.17, 95% CI: 0.13–0.23, *p* < 0.0001, ^h^: OR: 0.45, 95% CI: 0.26–0.80, *p* = 0.0055, ^i^: OR: 0.027, 95% CI: 0.003–0.236, *p* = 0.0002 (Fisher exact), ^j^: OR: 2.67, 95% CI: 1.50–4.72, *p* = 0.0006.

**Table 4 pathogens-12-01347-t004:** Distribution of cytological findings for the studied population by HPV and other than HPV STI status (the single SCC case was excluded).

		LBC *				
Status ***	Total n = 1197	NILM n = 800 (66.83%)	ASC-US n = 78 (6.52%)	LSIL n = 313 (26.15%)	HSIL n = 6 (0.5%)	*p*-Value
STIs (+) n (%)	404 (33.75)	231 (28.88) ^a^	26 (33.33)	145 (46.33) ^a^	2 (33.33)	<0.0001
STIs (−) n (%)	793 (66.25)	569 (71.13)	52 (66.67)	168 (53.67)	4 (66.67)	
HPV (+) n (%)	309 (25.81)	121 (15.13) ^b,d^	21 (26.92) ^c,e^	162 (51.76) ^d,e^	5 (83.33) ^b,c^	<0.0001
HPV (−) n (%)	888 (74.19)	679 (84.88)	57 (73.08)	151 (48.24)	1 (16.67)	
HPV+ STIs+ n (%)	153 (12.78)	57 (7.13)	7 (8.97)	88 (28.12)	1 (16.67)	0.1261
HPV+ STIs− n (%)	156 (13.03)	64 (8)	14 (17.95)	74 (23.64)	4 (66.67)	
HPV− STIs+ n (%)	251 (20.97)	174 (21.75) ^f^	19 (24.36)	57 (18.21) ^f^	1 (16.67)	0.0065
HPV− STIs− n (%)	637 (53.22)	505 (63.13)	38 (48.72)	94 (30.03)	0 (0)	
HPV HR+ n (%)	258 (21.55)	96 (12.00) ^g,h,i^	18 (23.08) ^h,j,k^	139 (44.41) ^g,j^	5 (83.33) ^i,k^	<0.0001
HPV HR− n (%)	939 (78.45)	704 (88.00)	60 (76.92)	174 (55.59)	1 (16.67)	
*Chlamydia T*. n (%)	30 (2.51)	14 (1.75)	2 (2.56)	14 (4.47)	0 (0)	0.0679 ^@^
*Mycoplasma H.* n (%)	88 (7.35)	51 (6.38)	11 (14.1)	26 (8.31)	0 (0)	0.0852 ^@^
*Mycoplasma G.* n (%)	9 (0.75)	3 (0.38)	5 (1.6)	5 (1.6)	0 (0)	0.1028 ^@^
*Ureaplasma* spp. n (%)	305 (25.48)	175 (21.88)	14 (17.95)	114 (36.42)	2 (33.33)	<0.0001 ^@^

* Includes women that had simultaneously valid results in STIs, HPV detection, and LBC; *** percentages in parentheses denote the percentage of the population within the cervical cytology diagnostic category being compliant with the denoted status. ^@^: Fisher exact test. Details of the statistical tests for the pairs that exhibited observed difference: ^a^: OR: 0.47, 95% CI: 0.36–0.62. *p* < 0.0001, ^b^: OR: 0.48, 95% CI: 0.28*0.83, *p* = 0.0069, ^c^: OR: 13.57, 95% CI: 1.50–123.04, *p* = 0.010 (Fisher exact), ^d^: OR: 0.17, 95% CI: 0.12–0.22, *p* < 0.0001, ^e^: OR: 2.91, 95% CI: 1.68–5.03, *p* < 0.0001, ^f^: OR: 0.57, 95% CI: 0.39–0.82, *p* = 0.0026, ^g^: OR: 0.17, 95% CI: 0.13–0.23, *p* < 0.0001, ^h^: OR: 0.45, 95% CI: 0.26–0.80, *p* = 0.0055, ^i^: OR: 0.027, 95% CI: 0.003–0.236, *p* = 0.0002 (Fisher exact), ^j^: OR: 2.67, 95% CI: 1.50–4.72, *p* = 0.0006, ^k^: OR: 16.67, 95% CI: 1.83–152.03, *p* = 0.0053 (Fisher exact).

## Data Availability

Data are available upon request from the corresponding author.

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
