# Peer review of "Cervical HPV Infections, Sexually Transmitted Bacterial Pathogens and Cytology Findings—A Molecular Epidemiology Study"

_pathogens, 2023, doi:10.3390/pathogens12111347_

Round 1

Reviewer 1 Report

Comments and Suggestions for Authors

Thanks for having me to review this manuscript. This study investigated the possible association of STI’s with cervical cytology aberrations and HPV genotyping results in a representative sample of predominantly young Greek women. The sample size is large, and the results is ample. The topic of this study is novel and bridges a significant gap. This study based on a nation-scale database, elucidated the concurrency of HPV infection and STI. References are appropriately selected.

The methodology is fine

My recommendation is revision.

Comment

-The introduction needs tremendous improvement. Three paragraphs are recommended.

-Table 4 should include the result of X2 test and odds ratio to improve the presentation of results.

-The conclusion looks more like a part of the discussionand should be refined. It should be based on the results, not on a lot of references.

Comments on the Quality of English Language

Good.

Author Response

Reviewer 1

Thanks for having me to review this manuscript. This study investigated the possible association of STI’s with cervical cytology aberrations and HPV genotyping results in a representative sample of predominantly young Greek women. The sample size is large, and the results is ample. The topic of this study is novel and bridges a significant gap. This study based on a nation-scale database, elucidated the concurrency of HPV infection and STI. References are appropriately selected.

The methodology is fine.

My recommendation is revision.

Comment 1: The introduction needs tremendous improvement. Three paragraphs are recommended.

Authors’ actions: Thank you for your comment. We have updated the introduction according to the journal’s instructions.

Comment 2: Table 4 should include the result of X2 test and odds ratio to improve the presentation of results.

Authors’ actions: We have added a new column in Table 4 to indicate the chi-square test p value. However, since odds ratio can be calculated only for two groups, the odds ratio, 95%CI and p values data were added under the table.

Comment 3: The Conclusion looks more like a part of the Discussion and should be refined. It should be based on the results, not on a lot of references.

Authors’ actions: Thank you for your point. To address this issue the Discussion section has been updated and one paragraph has been relocated in the Results section.

Reviewer 2 Report

Comments and Suggestions for Authors

This is a very interesting study of the correlation between HPV infection, which causes cervical cancer, and STIs in a large population.

At present, it is not known how much HPV infection is associated with other STIs, but the paper is highly useful as it also discusses previously reported cases of ureaplasma and chlamydia infection.

However, one point of concern is that it would be clinically useful to compare the risk of lesion development with the degree of STI infection (+ or ++ for bacterial infection), preferably as a prospective study. In the present study, for example, there are two possible cases of ASC-US, one due to HPV infection and the other due to inflammation; LgSIL, by definition, requires HPV infection. Are these effects likely to be present when focusing on cytology? Reconfirmation of pathological findings is requested.

Author Response

Reviewer 2

Comment 1: This is a very interesting study of the correlation between HPV infection, which causes cervical cancer, and STIs in a large population.

At present, it is not known how much HPV infection is associated with other STIs, but the paper is highly useful as it also discusses previously reported cases of ureaplasma and chlamydia infection.

Authors’ actions: Thank you for your comments regarding our study. A research in the literature indeed indicates there are no many studies investigating the concurrent existence of these bacterial and viral infections.

Comment 2: However, one point of concern is that it would be clinically useful to compare the risk of lesion development with the degree of STI infection (+ or ++ for bacterial infection), preferably as a prospective study. In the present study, for example, there are two possible cases of ASC-US, one due to HPV infection and the other due to inflammation; LgSIL, by definition, requires HPV infection. Are these effects likely to be present when focusing on cytology? Reconfirmation of pathological findings is requested.

Authorsactions: Thank you for your points; we really appreciate. Ideally a histologic confirmation (punch biopsy) would be highly valuable to document evidence of HPV infection. Similarly, utilizing another STI assay capable of providing information on the severity of the bacterial infection (Colony Forming Units, or a similar marker) would have been of value. However please consider that these caveats should be regarded in the context of a pragmatic multicenter prospective study, which embarked with limited human and financial resources. Additionally, in the “global” cervical biopsies for all patients scenario, ethical issues might arise regarding psychosomatic morbidity, especially for some of the LSIL lesions illustrating totally indolent colposcopic features. This point has been addressed in preliminary discussions with members of the Ethics Committee of one of the participating hospitals.

Reviewer 3 Report

Comments and Suggestions for Authors

The following points should be addressed to the Authors:

-          The aim of the study must be clarified at the end of introduction section. The Authors should also provide a research question based on the scientific evidence currently available in literature, improving the background information. At the moment, the aims of the study are unclear.

-          Line 115: Smoking status is not the only variable involved in carcinogenesis progress. Could the Authors specify each factor considered?

-          Line 136: The results, as number of cases and mean age, must be reported in the Results section. Moreover, mean age must include standard deviation and median age the IQR value.

-          Lines 138-143: This sentence is not clear. If the aim of the study was to evaluate the prevalence of HPV infection in women with previous or current STIs, why did the Authors decide to also consider participants with missing information?

-          Line 195, please provide the total number of involved participants.

-          Could the Authors specify how many participants had cytology and HPV-DNA results? Which is the prevalence of HPV infection based on cytology classification? This important information should be included in the text (not only in Supplementary).

-          Although the interesting comparison with national epidemiological data in Discussion, it is important to extend the comparison with international literature. In this case, I’d suggest you consider recent studies in similar context, as: 10.3390/ijerph19020693 for prevalence of HPV infection in ASCUS-positive women and 10.3390/ijerph16183354 for prevalence of Chlamydia infection in Hr-HPV-positive population.

Minor revisions:

-          Add “introduction” at the beginning of the manuscript.

-          Please, change “LgSIL” in “LSIL”, and “HgSIL” in “HSIl”.

-          Line 274-284: Please, insert in the manuscript the reference to Supplementary materials.

-          Please, remove the paragraph in discussion section or, at least, follow the same paragraphs reported in results section.

-          The References list must be uniformed, following the journal’s instructions.

Comments on the Quality of English Language

Moderate editing of English language is required.

Author Response

Reviewer 3

The following points should be addressed to the Authors:

Comment 1: The aim of the study must be clarified at the end of introduction section. The Authors should also provide a research question based on the scientific evidence currently available in literature, improving the background information. At the moment, the aims of the study are unclear.

Authors’ actions: Thank you for your point. We have made changes to clarify the scopes of the study. Please note that this is a pragmatic study, i.e. in real world conditions an attempt to investigate the presence and co-existence of different pathogens in a population attending different clinics in the context of cervical screening.

Comment 2: Line 115: Smoking status is not the only variable involved in carcinogenesis progress. Could the Authors specify each factor considered?

Authors’ actions: Thank you for your point. Indeed, the reason that we have not included smoking data in our results and analysis, despite them being available, was the substantial use of alternative tobacco products (e-cigarette & vaping) among the studied population. We have added a clear statement in the limitations of the study section.

Comment 3: Line 136: The results, as number of cases and mean age, must be reported in the Results section. Moreover, mean age must include standard deviation and median age the IQR value.

Authors’ actions: We do apologize; it was our omission and fault using “median” values instead of “mean”. We have now corrected and transferred to the results section.

Comment 4: Lines 138-143: This sentence is not clear. If the aim of the study was to evaluate the prevalence of HPV infection in women with previous or current STIs, why did the Authors decide to also consider participants with missing information?

Authors’ actions: In the Venn diagram, the intersection of the two circles illustrates the number of valid results for both set. We embarked in this methodology which has been previously implemented by other authors to achieve a larger sample. To assess the methodology for missing data, we previously performed an evaluation of the effect of imputations in the final estimations

Comment 5: Line 195, please provide the total number of involved participants.

Authors’ actions: The total number of involved participants is 2268. A relevant sentence was transferred from the materials section here and indicates the actual number of eligible women.

Comment 6: Could the authors specify how many participants had cytology and HPV-DNA results? Which is the prevalence of HPV infection based on cytology classification? This important information should be included in the text (not only in Supplementary).

Authors’ actions: The Venn diagram is now corrected and made more informative, the lines within the intersecting circles show the actual number of women that have two (or three) valid results available simultaneously. The number of women with valid HPV-DNA and cytology is 1198+13=1211.

Comment 7: Although the interesting comparison with national epidemiological data in Discussion, it is important to extend the comparison with international literature. In this case, I’d suggest you consider recent studies in similar context, as: 10.3390/ijerph19020693 for prevalence of HPV infection in ASCUS-positive women and 10.3390/ijerph16183354 for prevalence of Chlamydia infection in Hr-HPV-positive population.

Authors’ actions: Thank you for your advice to enrich our study. We have added 1 paragraph in the discussion section referring on your useful suggested publications.

Minor revisions:

Comment 8: Add “introduction” at the beginning of the manuscript.

Authors’ actions: We have added “Introduction” in the begging of the manuscript. Thank you.

Comment 9: Please, change “LgSIL” in “LSIL”, and “HgSIL” in “HSIL”.

Authors’ actions: We have changed “LgSIL & HgSIL” to LSIL & HSIL throughout the manuscript accordingly, including the labels of the figures.

Comment 10: Line 274-284: Please, insert in the manuscript the reference to Supplementary materials.

Authors’ actions: A Supplementary table has been already mentioned and included within the main text of the manuscript as Table S1 (Lines 321, 324, 331, 334, 337 and 341).

Comment 11: Please, remove the paragraph in discussion section or, at least, follow the same paragraphs reported in results section.

Authors’ actions: We have adhered to your advice and the paragraph has been placed in the “Results” section instead of “Discussion”.

Comment 12:  The References list must be uniformed, following the journal’s instructions.

Authors’ actions: The Reference list is now updated according to the MDPI publishing group instructions. Thank you!

Reviewer 4 Report

Comments and Suggestions for Authors

The investigators conducted a multicenter molecular epidemiology study in a cohort of 2268 Greek women to study the possible association of sexually transmitted bacterial infection and HPV infection with cervical cytology aberrations. This study is significant to the field, given its large sample size and the scarcity of such comprehensive investigations in this regional context. HPV and STI pose a reciprocal risk and this study corroborates so. It concludes that women testing positive both for high-risk as well as low-risk HPV genotypes are more prone to test positive for a bacterial STI as compared to those testing positive either for a high-risk or a low-risk HPV genotype alone. This study provides supporting data to the common observation that co-infections lead to an increased risk of cervical cytology aberrations and highlights the complex interplay between viral and bacterial pathogens in cervical health.

Below are a few comments that should be incorporated into the manuscript and revised before being considered for publication.

  1. Line 62: The authors note, "Among 1361 individuals with valid results both for bacterial STI’s and HPV detection, women with a HPV positive sample were more likely to harbor a bacterial STIs”. According to Figure 1 (Page 10), and its footnote (line 149) there were 1371 study participants with valid HPV and STI detection assay results. The authors should justify the exclusion of the remaining 10 participants from the analysis.
  2. Line 234: “Out of 58 women in the younger age group (≤20 years old) for whom a valid STI and an HPV test were both available...” Data from Table 3 (page 7) shows that the analysis is done only for n=1344 participants while Figure 1 (page 4) suggests there were as many as 1371 participants with valid HPV and STI detection assay results. This mandates clarification.
  3. Line 265: “...  however for the HPV negative sub-group, bacterial STI positivity rate was higher in the LgSIL group than in NILM cases (37.75% vs. 240 25.63%, p<0.05, see Table 4)”. This data cannot be verified.

Author Response

Reviewer 4

The investigators conducted a multicenter molecular epidemiology study in a cohort of 2268 Greek women to study the possible association of sexually transmitted bacterial infection and HPV infection with cervical cytology aberrations. This study is significant to the field, given its large sample size and the scarcity of such comprehensive investigations in this regional context. HPV and STI pose a reciprocal risk and this study corroborates so. It concludes that women testing positive both for high-risk as well as low-risk HPV genotypes are more prone to test positive for a bacterial STI as compared to those testing positive either for a high-risk or a low-risk HPV genotype alone. This study provides supporting data to the common observation that co-infections lead to an increased risk of cervical cytology aberrations and highlights the complex interplay between viral and bacterial pathogens in cervical health.

Below are a few comments that should be incorporated into the manuscript and revised before being considered for publication.

Comment 1: Line 62: The authors note, "Among 1361 individuals with valid results both for bacterial STI’s and HPV detection, women with a HPV positive sample were more likely to harbor a bacterial STIs”. According to Figure 1 (Page 10), and its footnote (line 149) there were 1371 study participants with valid HPV and STI detection assay results. The authors should justify the exclusion of the remaining 10 participants from the analysis.

Authors’ actions: Thank you for noticing. This was a typographic mistake; the correct number is 1371. Moreover, we have altered the Venn diagram to show the women that have simultaneous valid tests results available, hoping this clarifies better the data.

Comment 2: Line 234: “Out of 58 women in the younger age group (≤20 years old) for whom a valid STI and an HPV test were both available...” Data from Table 3 (page 7) shows that the analysis is done only for n=1344 participants while Figure 1 (page 4) suggests there were as many as 1371 participants with valid HPV and STI detection assay results. This mandates clarification.

Authors’ actions: In this sentence we only focused on the subgroup of women within the first, youngest age group (≤20 years old). We have rephrased this sentence for clarity. Thank you!

Comment 3: Line 265: “...however for the HPV negative sub-group, bacterial STI positivity rate was higher in the LgSIL group than in NILM cases (37.75% vs. 240 25.63%, p<0.05, see Table 4)”. This data cannot be verified.

Authors’ actions: In the revised version the odds ratios and relevant confidence intervals have been included; this sentence was changed accordingly.

Round 2

Reviewer 2 Report

Comments and Suggestions for Authors

Thank you very much for your revisions. Now I don’t have any concerns regarding the manuscript.

Reviewer 3 Report

Comments and Suggestions for Authors

I thank the Authors for positively accepting my suggestions and making the changes in the text.

Comments on the Quality of English Language

Minor editing